

# Airborne DOAS retrievals of methane, carbon dioxide, and water vapor concentrations at high spatial resolution: application to AVIRIS-NG

Andrew K. Thorpe [1], Christian Frankenberg [2,1], David R. Thompson [1], Riley M. Duren [1], Andrew D. Aubrey [1], Brian B. Bue [1], Robert O. Green [1], Konstantin Gerilowski [3], Thomas Krings [3], Jakob Borchardt [3], Eric A. Kort [4], Colm Sweeney [5], Stephen Conley [6,7], Dar A. Roberts [8], and Philip E. Dennison [9]

[1]Jet Propulsion Laboratory, California Institute of Technology, Pasadena, California, United States
[2]California Institute of Technology, Pasadena, California, United States
[3]Institute of Environmental Physics (IUP), University of Bremen, Bremen, Germany
[4]University of Michigan, Ann Arbor, United States
[5]Cooperative Institute for Research in Environmental Sciences, University of Colorado, Boulder, CO, United States
[6]Global Monitoring Division, NOAA Earth System Research Laboratory, Boulder, Colorado, United States
[7]Scientific Aviation, 3335 Airport Road, Boulder, Colorado, United States
[8]University of California, Santa Barbara, Santa Barbara, California, United States
[9]University of Utah, Salt Lake City, Utah, United States

*Correspondence to:* Andrew K. Thorpe (Andrew.K.Thorpe@jpl.nasa.gov)

**Abstract.** At local scales, emissions of methane and carbon dioxide are highly uncertain. Localized sources of both trace gases can create strong local gradients in its columnar abundance, which can be discerned using absorption spectroscopy at high spatial resolution. In a previous study, more than 250 methane plumes were observed in the San Juan Basin near Four Corners during April 2015 using the next generation Airborne Visible/Infrared Imaging Spectrometer (AVIRIS-NG) and a linearized

matched filter. For the first time, we apply the Iterative Maximum a Posteriori Differential Optical Absorption Spectroscopy (IMAP-DOAS) method to AVIRIS-NG data and generate gas concentration maps for methane, carbon dioxide, and water vapor plumes. This demonstrates a comprehensive greenhouse gas monitoring capability that targets methane and carbon dioxide, the two dominant anthropogenic climate-forcing agents. Water vapor results indicate the ability of these retrievals to distinguish between methane and water vapor despite spectral mixing in the short wave infrared. We focus on selected cases

from anthropogenic and natural sources, including emissions from mine ventilation shafts, a gas processing plant, tank, pipeline leak, and natural seep. In addition, carbon dioxide emissions were mapped from the flue-gas stacks of two coal-fired power plants and a water vapor plume was observed from the cooling towers of one power plant. Observed plumes were consistent with known and suspected emission sources verified by the true color AVIRIS-NG scenes and higher resolution Google Earth imagery. Real time detection and geolocation of methane plumes by AVIRIS-NG provided unambiguous identification of

individual emission source locations and communication to a ground team for rapid follow up. This permitted verification of a number of methane emission sources using a thermal camera, including a tank and buried natural gas pipeline.



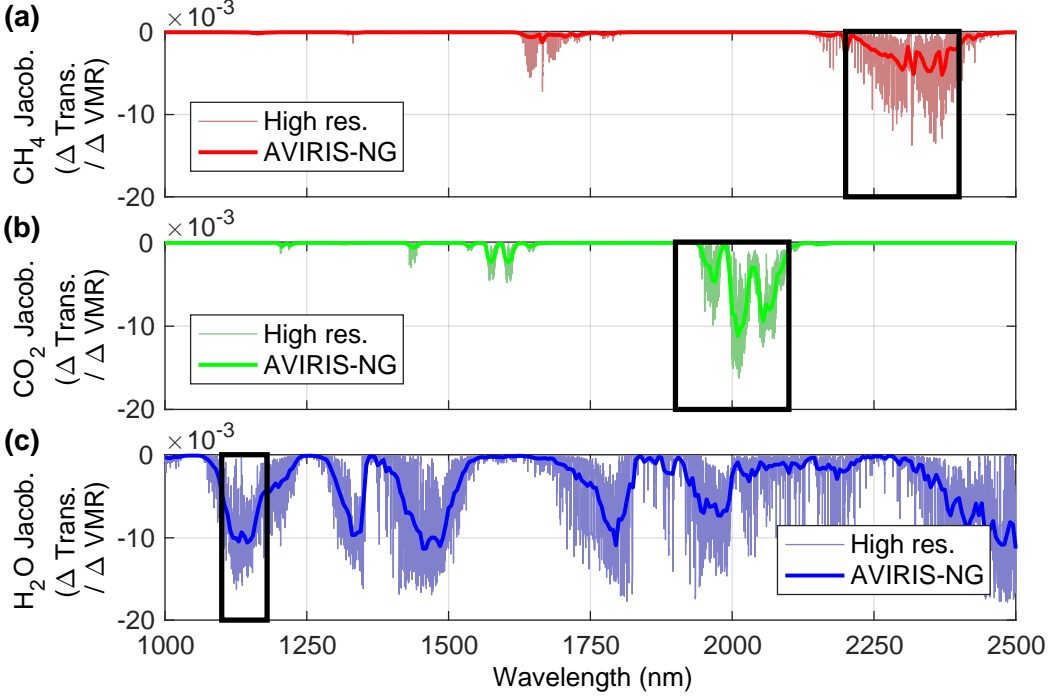

**Figure 1.** High resolution gas Jacobians plotted in lighter colors and for AVIRIS-NG (5 nm spectral resolution and sampling) for (a) $CH_4$ (red), (b) $CO_2$ (green), and (c) $H_2O$ (blue). These examples were calculated for a 5 % change in $CH_4$ (red), (b) $CO_2$ (green), and (c) $H_2O$ over the total column. AVIRIS-NG retrieval windows are indicated by the black outlines.

# 1 Introduction

It is important to better understand the processes controlling changes in atmospheric methane ($CH_4$) and carbon dioxide ($CO_2$), the two dominant anthropogenic climate-forcing agents. $CH_4$ and $CO_2$ contribute approximately 17 % and 64 % of the total radiative forcing attributed to anthropogenic greenhouse gases and halocarbons (Myhre et al., 2013). The atmospheric growth rates are strongly influenced by anthropogenic emissions of $CH_4$ and dominated by fossil fuel $CO_2$ emissions. Anthropogenic $CH_4$ sources were estimated to contribute 10.6 % of the total 2014 anthropogenic emissions of the United States, with major sources including natural gas systems (2.6 %), enteric fermentation (2.4 %), landfills (2.2 %), petroleum systems (1.0 %), coal mining (1.0 %) (EPA, 2016a). $CH_4$ is a precursor for tropospheric ozone and is strongly linked with co-emitted reactive trace gases that are the focus of air quality mitigation policies. U.S. anthropogenic $CO_2$ sources make up 81 % of the total anthropogenic emissions and are dominated by fossil fuel combustion, including electricity generation (30 %), transportation (25 %), and industrial emissions (12 %) (EPA, 2016a). U.S. emissions of both gases are projected to increase (EIA, 2013) and a number of studies have suggested that EPA bottom up emission inventories are underestimated for $CH_4$ (Miller et al., 2013; Wecht et al., 2014; Turner et al., 2015). U.S. fossil fuel $CO_2$ emissions are better constrained through existing inventories of fossil



fuel sales and combustion, however, global uncertainties are growing with the rise of a number of large, developing countries where emissions information is not readily available (NRC, 2010; Ballantyne et al., 2015; Ciais et al., 2015).

There remains uncertainty regarding the sources and sinks of atmospheric $CH_4$, as reflected by the ongoing scientific discussion on both the hiatus in the atmospheric growth rate in the early 21st century as well as the unexpected rise starting in 2007 (Nisbet et al., 2014). Further, regional top-down emissions estimates cannot discriminate source categories and thereby

attribute fluxes to specific processes or sources. Uncertainty in anthropogenic $CH_4$ emissions is large at multiple scales and process attribution remains challenging because emissions originate from biological processes, venting, and leaks (Kirschke et al., 2013; Schwietzke et al., 2016; Schaefer et al., 2016).

Recent studies suggest that the majority of $CH_4$ emissions from oil and gas supply chains are caused by a number of super-emitters, which could explain underestimates in bottom-up inventories (Zavala-Araiza et al., 2015; Lyon et al., 2015; Brandt

et al., 2014, 2016). The ability to identify emission sources offers the potential to constrain regional greenhouse gas budgets and improve partitioning between anthropogenic and natural emission sources. Although $CH_4$ has a short atmospheric lifetime (about 9 years), it has a very high Global Warming Potential (GWP) which is 86 times greater than $CO_2$ on a 20 year time scale (Myhre et al., 2013). This means that even small amounts of emissions reduction will result in large reductions in the overall atmospheric radiative forcing.

Driving surveys using in situ instruments have been used to identify $CH_4$ emission sources in major U.S. metropolitan areas like the Los Angeles basin (Hopkins et al., 2016), Boston (Phillips et al., 2013), and Washington, D.C. (Jackson et al., 2014) as well as to measure fluxes (Rella et al., 2015). Recently, ground-based thermal imaging systems have also been used to identify $CH_4$ emissions (Johnson et al., 2015; Galfalk et al., 2016). However, these methods require comprehensive sampling techniques, are time consuming, and can be limited to regions with sufficient road access. In situ airborne measurements offer

the potential for increased coverage and have been used for U.S. regional $CH_4$ flux estimates using mass balance approaches for the Uintah Basin in northeastern Utah (Karion et al., 2013), the Marcellus formation in southwestern Pennsylvania (Caulton et al., 2014), and the Barnett Shale formation in Texas (Smith et al., 2015; Lavoie et al., 2015). These measurements reflect gas concentrations at the flight altitude and these studies are designed to estimate aggregate emissions for large regions rather than identifying individual emissions sources.

More recently, in situ airborne measurements using a chemically-instrumented Mooney aircraft have been used to estimate fluxes from known sources like the Aliso Canyon leak (Conley et al., 2016) and for a number of sources identified by imaging spectrometers in the Four Corners region (Frankenberg et al., 2016). This method samples the atmosphere directly at the flight path altitude and can measure multiple gas species. The Methane Airborne MAPper (MAMAP) spectrometer (Gerilowski

et al., 2011) has also been used to measure elevated $CH_4$ and $CO_2$ column abundances to quantify emissions from a coal mine ventilation shaft (Krings et al., 2013), power plants (Krings et al., 2011), and a landfill (Krautwurst et al., 2016). MAMAP is a non-imaging spectrometer with a small field of view limited to flying transects across gas plumes rather than quickly mapping their morphology and extent on small scales. Both instruments are better suited for investigating either known emission sources or identifying larger regional emissions as opposed to individual sources.





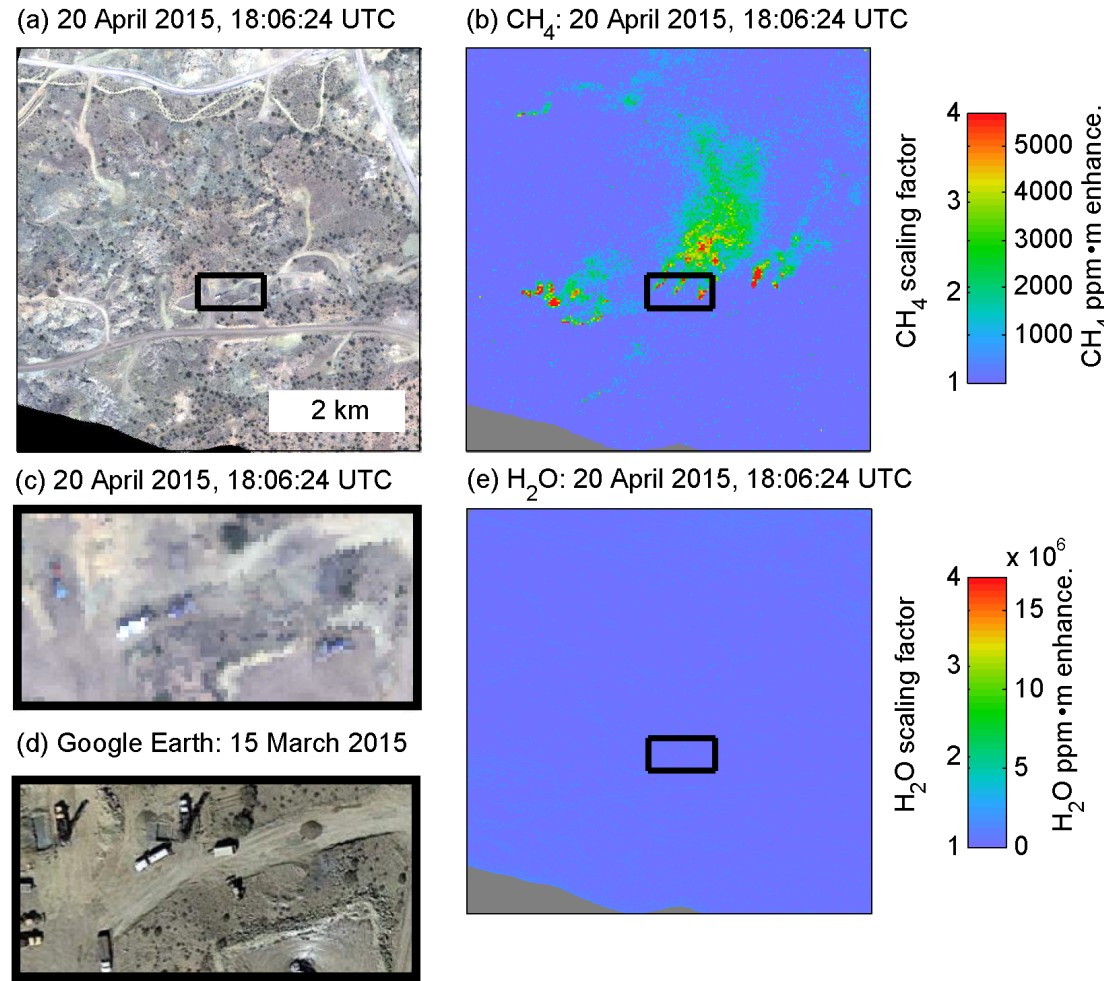

**Figure 2.** (a) AVIRIS-NG true color image subset. (b) A number of $CH_4$ plumes are clearly visible with maximum enhancements in excess of 5,000 ppm·m. (c) Close up of AVIRIS-NG true color image shown by black outline in (a). (d) Higher resolution Google Earth imagery for same area reveals drilling rigs at an active underground coal mine, suggesting the origin of these plumes are mine workings ventilation shafts. (e) $H_2O$ retrieval does not indicate enhancements. For all images, north is up.

## 2 Airborne imaging spectrometers

Airborne imaging spectrometers like the Airborne Visible/Infrared Imaging Spectrometer (AVIRIS) (Green et al., 1998) and the next generation instrument AVIRIS-NG (Hamlin et al., 2011) can map large regions while providing the spatial resolution required to identify individual emissions within scenes. While not originally designed for mapping emissions, these instruments measure the 0.38 to 2.5 $\mu$m range that includes many gas absorption features (Figure 1). This has permitted quantitative retrievals of $CH_4$ using AVIRIS (approximately 10 nm spectral resolution and sampling) for marine seeps (Roberts et al.,





2010; Thorpe et al., 2014). Water vapor retrievals have been demonstrated with AVIRIS (Gao and Goetz, 1990; Thompson et al., 2015a) mainly for atmospheric correction and reflectance retrievals. However, AVIRIS water vapor retrievals have also been used to measure plant transpiration, demonstrating potential application to the fields of ecology and meteorology (Ogunjemiyo et al., 2004).

AVIRIS has been used for high resolution mapping of $CO_2$ plumes from industrial sources (Dennison et al., 2013) and wildfires (Marion et al., 2004; Deschamps et al., 2011). More recently, AVIRIS-NG (approximately 5 nm spectral resolution and sampling) has surveyed large regions to identify $CH_4$ emissions associated with oil production (Thompson et al., 2015b), gas extraction (Frankenberg et al., 2016), hydraulic fracturing (Aubrey et al., 2015), and a landfill (Krautwurst et al., 2016). This is possible due to a 34° field of view, which results in an image swath of 1.8 km when flying at 3 km above ground level

(AGL).

Airborne imaging spectrometers that operate in the thermal infrared, such as the Mako and HyTES instruments (Tratt et al., 2014; Hulley et al., 2016), have also been used for mapping $CH_4$ plumes. However, the altitude of maximum sensitivity varies with environmental conditions like thermal contrast (Kuai et al., 2016), which can make plumes difficult to detect and quantify, and sensitivity to near surface emissions decreases with flight altitude, which can limit ground coverage. Because AVIRIS and

AVIRIS-NG measure reflected solar radiation in the short wave infrared, $CH_4$ retrieval sensitivity is impacted only slightly by flight altitude due to additional gas attenuation along the optical path. However, at higher flight altitude and coarser spatial resolution a gas enhancement is diluted over a larger image pixel, thereby decreasing instrument sensitivity. The ability to fly high results in more efficient flight campaigns due to improved ground coverage. For example, AVIRIS-NG consistently observed plumes for a $CH_4$ controlled release experiment for all altitudes flown (up to 3.8 km AGL) and AVIRIS has observed $CH_4$ plumes flying at 8.9 km AGL (Thorpe et al., 2014). AVIRIS has also mapped $CH_4$ plumes over multiple days from the Aliso Canyon leak by flying 6.6 km AGL, resulting in an image swath approximately 4.0 km wide (Thompson et al., 2016). This also offers the potential for space-based detection of emission sources, like the observed $CH_4$ plume from Aliso Canyon using the orbital Hyperion imaging spectrometer (Thompson et al., 2016).

In a previous study (Thorpe et al., 2014), the Iterative Maximum a Posteriori Differential Optical Absorption Spectroscopy (IMAP-DOAS) retrieval was applied to AVIRIS for quantitative mapping of $CH_4$ from natural and anthropogenic sources. In

this study, the application of IMAP-DOAS has been expanded for use with AVIRIS-NG for multiple gas species, including $CH_4$, $CO_2$, and $H_2O$. We present results from AVIRIS-NG data acquired in New Mexico and Colorado, including from a flight campaign in the San Juan Basin near Four Corners. We will present results for a number of sources, including $CH_4$ from mine ventilation shafts, a gas processing plant, tank, pipeline leak, and natural seep, as well as $CO_2$ and $H_2O$ plumes associated with power plants.

**3   Study sites and AVIRIS-NG data**

Space-based observations collected by the SCanning Imaging Absorption SpectroMeter for Atmospheric CHartographY (SCIA-MACHY) instument (Bovensmann et al., 1999) showed $CH_4$ enhancements in the Four Corners region (Kort et al., 2014). This





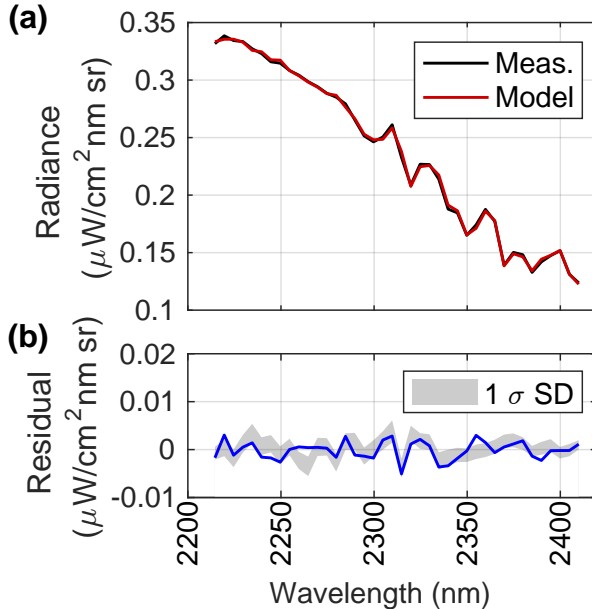

**Figure 3.** (a) AVIRIS-NG measured and modeled radiance for one image pixel within the $CH_4$ plume used for the $CH_4$ retrieval (see Figure 2b). (b) The residual is plotted with 1 $\sigma$ standard deviation boundary calculated from residuals for the entire scene.

made for an ideal location for follow up surveys using AVIRIS-NG to identify individual emission sources. During the flight campaign, the AVIRIS-NG instrument was equipped with a real time $CH_4$ mapping capability using a waterfall display mon-

itored by the instrument operator. Observed $CH_4$ plumes were overlaid on a true color image displaying location information as well as the maximum $CH_4$ enhancement (Thompson et al., 2015b). This permitted adaptive survey strategies to investigate observed plumes and the ability to send images of the plume with accurate locations to a ground crew for subsequent follow up. A Xenics Onca-VLWIR-MCT-384 thermal imaging camera with a Spectrogon optical filter centered at 7.746 $\mu$m was used by the ground crew to verify a number of plumes observed in real time by AVIRIS-NG.

5     Located in New Mexico and Colorado, the San Juan Basin produces natural gas from sandstone, coal bed $CH_4$ and shale formations and is the fourth largest U.S. gas field when it comes to total production (EIA, 2015). During a five day campaign in April 2015, AVIRIS-NG targeted an area corresponding to the highest $CH_4$ enhancements observed with SCIAMACHY (Frankenberg et al., 2016). A 2,500 km$^2$ area was covered in approximately two days (9.2 flight hours) flying at 3 km above ground level, resulting in scenes with an image swath of around 1.8 km and a ground resolution of 3 m. The remaining flight

10  days were used for additional follow up flights and some repeat observations, sometimes at lower flight altitudes. During the campaign, a number of potential $CH_4$ emission sources were targeted, including infrastructure associated with natural gas production like wellpads, tanks, and gas processing plants, a coal mine, and natural coal bed $CH_4$ seeps. While the flight campaign focused on $CH_4$ sources, the coal-fired San Juan power generating station was also flown as a potential $CO_2$ emission source.





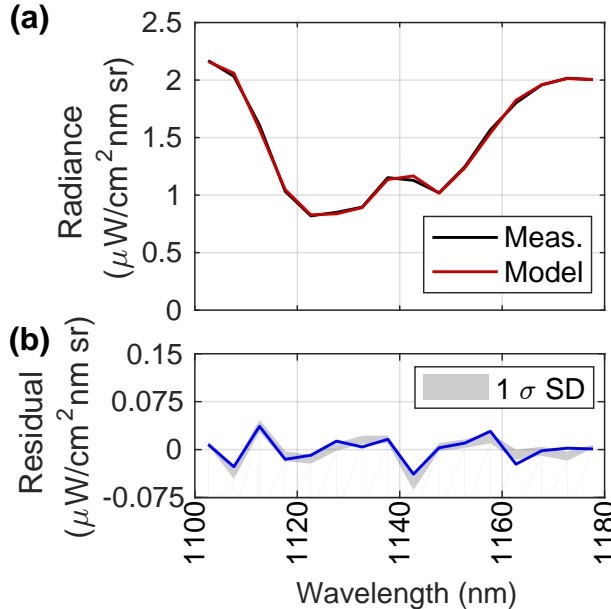

**Figure 4.** (a) AVIRIS-NG measured and modeled radiance for one image pixel within the $CH_4$ plume used for the $H_2O$ retrieval (see Figure 2e). (b) The residual is plotted with 1 $\sigma$ standard deviation boundary calculated from residuals for the entire scene.

## 4   IMAP-DOAS retrievals

A detailed description of the IMAP-DOAS retrieval for AVIRIS can be found in Thorpe et al. (2014). Gas retrievals were performed on orthocorrected radiance data. Atmospheric profiles were generated by updating prior gas profiles from the U.S. standard atmosphere obtained from the radiative transfer models LOWTRAN/MODTRAN (Kneizys et al., 1996) using volume

mixing ratios (VMR) from the NOAA Mauna Loa station, United States (NOAA, 2015). Temperature, pressure, and water vapor VMR profiles representative of the time period of the flight campaign were acquired from the National Centers for Environmental Prediction/National Center for Atmospheric Research (NCEP/NCAR) Reanalysis project (Kalnay et al., 1996). Spectral parameters for $CH_4$, $CO_2$, $H_2O$, and $N_2O$ were used from the HITRAN database (Rothman et al., 2009) and a classical Voigt spectral line-shape was used to calculate vertical optical densities for fourteen atmospheric layers that spanned sea level

to the top of the atmosphere.

   Above the aircraft, vertical optical densities were combined and an air mass factor (AMF) was calculated to account for one way transmission. Vertical optical densities below the aircraft were also combined with an AMF reflecting two way transmission. This resulted in a two layer atmospheric model that speeds up the retrieval and incorporates the ground elevation and flight altitude for each AVIRIS-NG scene. The two layer model was used to model reflected solar radiation perturbed by

the absorbing species $CH_4$, $CO_2$, $H_2O$, and $N_2O$. Three retrieval windows were used, each targeting the primary gas of interest. $CH_4$ retrievals were performed between 2,215 and 2,410 nm (Figure  1) and included fits for $H_2O$ and $N_2O$. Gas Jacobians



that reflect changes in absorption due to the absorbing species $CH_4$, $CO_2$, $H_2O$ are shown in Figure 1. Because $N_2O$ has weak absorption features, these Jacobians are not shown in Figure 1. Between 1,904 and 2,099 nm $CO_2$ retrievals included $H_2O$ and $N_2O$, while $H_2O$ retrievals between 1,103 and 1,178 nm also included $CO_2$ and $N_2O$. Therefore, the state vector ($\boldsymbol{x}_n$) for

each retrieval window has 6 entries (three gases for two atmospheric layers). Modeled radiance at high spectral resolution was calculated for each wavelength with a forward radiative transfer model using the following equation

$$\boldsymbol{F}^{\mathrm{hr}}(\boldsymbol{x}_i) = \boldsymbol{I}_0^{\mathrm{hr}} \cdot \exp\left(-\sum_{n=1}^{6} \boldsymbol{A}_n \cdot \boldsymbol{\tau}_n^{\mathrm{ref}} \cdot \boldsymbol{x}_{n,i}\right) \cdot \sum_{i=0}^{k} a_k \lambda^k, \tag{1}$$

where $\boldsymbol{F}^{\mathrm{hr}}(\boldsymbol{x}_i)$ is the forward modeled radiance at the $i$th iteration of the state vector, $\boldsymbol{I}_0^{\mathrm{hr}}$ is the incident intensity, a solar transmission spectrum (Geoffrey Toon, personal communication, 2013), $\boldsymbol{A}_n$ is the air mass factor (AMF) for each $n$ number

of atmospheric state vector elements, $\boldsymbol{\tau}_n^{\mathrm{ref}}$ is the reference vertical optical density for each n number of atmospheric state vector elements (including optical densities of the three absorbing species, $\boldsymbol{x}_{n,i}$ is the trace gas related state vector at the $i$th iteration, which scales the prior optical densities of each of the absorbing species in each $n$ layer (six rows, three gases for two atmospheric layers), $a_k$ are polynomial coefficients to account for low-frequency spectral variations.

The state vector contains the spectral shift (not shown here) and a low order polynomial function ($a_k$) to account for the

broadband variability in surface albedo (see Frankenberg et al. (2005). The high resolution modeled radiance is convolved using the the instrument line shape function and sampled to the center wavelengths for each AVIRIS-NG spectral band, resulting in a lower resolution modeled radiance at the $i$th iteration of the state vector $\boldsymbol{F}^{\mathrm{lr}}(\boldsymbol{x}_i)$, calculated using a known $\boldsymbol{\tau}_n^{\mathrm{ref}}$ scaled by $\boldsymbol{x}_{n,i}$.

A Jacobian Matrix is calculated for each iteration i, where each column represents the derivate vector of the sensor radiance with respect to each element of the state vector ($\boldsymbol{x}_i$).

$$\mathbf{K}_i = \left.\frac{\partial \boldsymbol{F}^{\mathrm{lr}}(\boldsymbol{x})}{\partial \boldsymbol{x}}\right|_{\boldsymbol{x}_i}. \tag{2}$$

The state vector at the $i$th iteration can be optimized as follows (Rodgers, 2000)

$$\boldsymbol{x}_{i+1} = \boldsymbol{x}_{\mathrm{a}} + \left(\mathbf{K}_i^T \mathbf{S}_\varepsilon^{-1} \mathbf{K}_i + \mathbf{S}_{\mathrm{a}}^{-1}\right)^{-1} \mathbf{K}_i^T \mathbf{S}_\varepsilon^{-1}$$
$$\cdot \left[\boldsymbol{y} - \boldsymbol{F}^{\mathrm{lr}}(\boldsymbol{x}_i) + \mathbf{K}_i(\boldsymbol{x}_i - \boldsymbol{x}_{\mathrm{a}})\right], \tag{3}$$

where $\boldsymbol{x}_{\mathrm{a}}$ is the a priori state vector (six rows), $\boldsymbol{x}_i$ is the state vector at the $i$th iteration (six rows), $\mathbf{S}_\varepsilon$ is the error covariance matrix, $\mathbf{S}_{\mathrm{a}}$ is the a priori covariance matrix, $\boldsymbol{y}$ is the measured AVIRIS-NG radiance, $\boldsymbol{F}^{\mathrm{lr}}(\boldsymbol{x}_i)$ is the forward model evaluated at $\boldsymbol{x}_i$, and $\mathbf{K}_i$ is the Jacobian of the forward model at $\boldsymbol{x}_i$.

The retrieval optimizes a scaling factor relative to the a priori profile. The a priori scaling factor is set to one as an initial guess for each gas in the two layers, while the a priori covariance matrix was set to constrain the fit to the atmospheric layer beneath the aircraft where high variance is expected. To do so, very small prior covariances were set for the uppermost layer (above the aircraft). Because the observed plumes are not expected to extend above the AVIRIS-NG flight altitude, this assumption is reasonable. Gas concentrations were calculated in ppm·m by multiplying the gas state vector at the last iteration (gas scaling





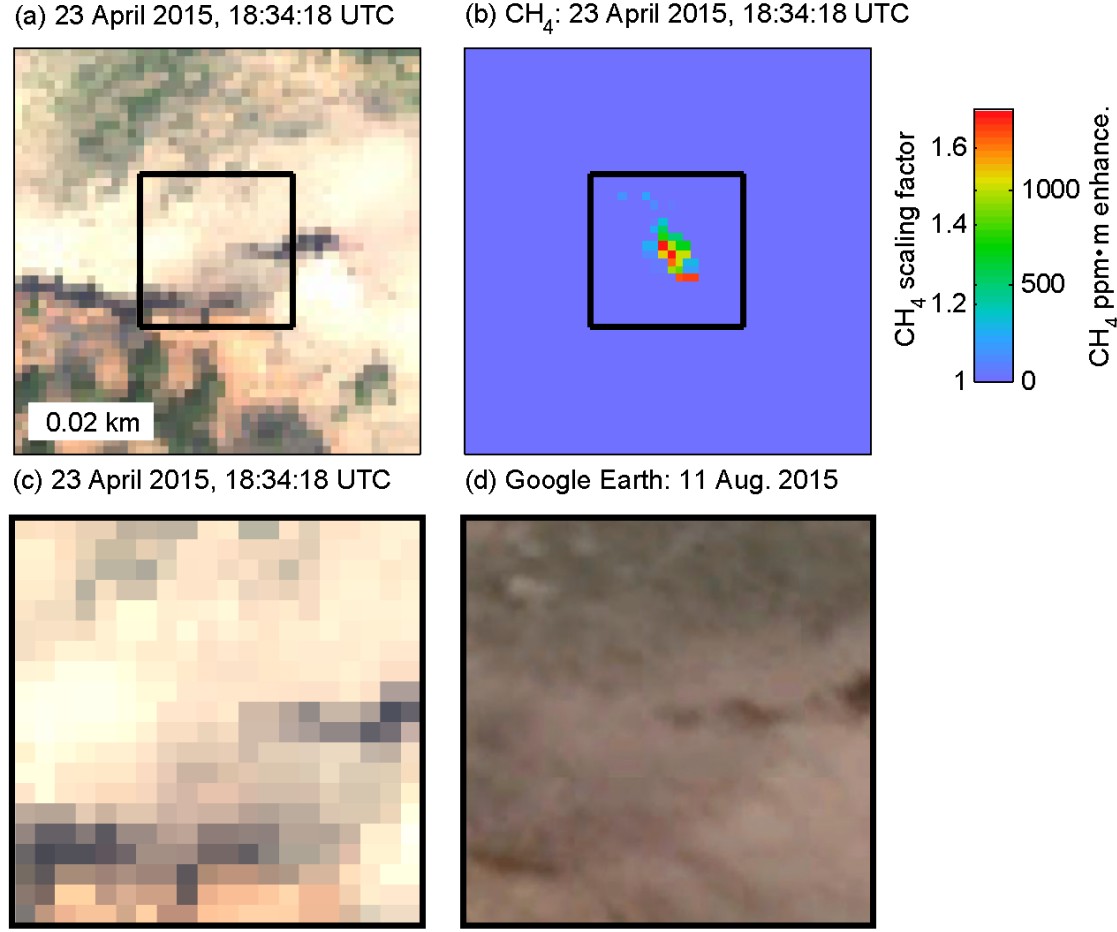

**Figure 5.** (a) AVIRIS-NG true color image subset. (b) A small $CH_4$ plume is visible from a confirmed geological source at Moving Mountain near Durango, Colorado. (c) Close up of AVIRIS-NG true color image. (d) Higher resolution Google Earth imagery provides additional spatial context. For all images, north is up.

factor) by the VMR for the lowest layer of the reference atmosphere and the distance between the aircraft and the ground. In subsequent figures, color bars will indicate the scaling factors as well as gas enhancements relative to background, which were calculated by subtracting the retrieved gas concentration from the background concentration for the lowest layer of the reference atmosphere.





## 5   Results

### 5.1   CH$_4$ emissions from natural gas sector

While AVIRIS-NG identified over 250 CH$_4$ plumes during the Four Corners flight campaign (Frankenberg et al., 2016), IMAP-DOAS retrievals for only a few examples will be presented here. The first example from a 20 April 2015 flight at 1.1 km AGL (Figure 2b) is made up of at least 10 plumes with maximum enhancements in excess of 5,000 ppm·m, which is equivalent to a concentration of 0.5 % in a 1 m thick layer or roughly an XCH4 (dry air column-averaged mole fraction) enhancement of around 500 ppb that is almost 25 % of a total background column. Results from the H$_2$O retrieval (Figure 2e) do not indicate enhancements collocated with CH$_4$ plumes. The true color image subset in Figure 2a reveals a few dirt roads, however, the close up of the AVIRIS-NG scene indicated by the black boxes in Figure 2a, b indicates some visible infrastructure that is difficult to interpret at the 1 m AVIRIS-NG pixel resolution (Figure 2c).

In Figure 2d, Google Earth imagery for the same area provides improved spatial resolution and reveals what appears to be drilling rigs at an active underground coal mine on 15 March 2015, suggesting the origin of these plumes are mine workings ventilation shafts. Frankenberg et al. (2016) estimated an aggregate flux of 2,236 kg hr$^{-1}$ for these plumes. Measured and modeled radiance is shown for one image pixel within the CH$_4$ plume for the CH$_4$ retrieval fitting window (Figure 3a) and for the H$_2$O retrieval (Figure 4a). For both examples, the residuals are also plotted (Figure 3b, Figure 4b) in addition to the 1 $\sigma$ standard deviation boundary calculated from residuals for the entire scene.

Additional examples are presented in Appendix A, including from another 20 April 2015 flight at 1.4 km AGL that results in a 1.2 m resolution (Figure 8b). Multiple CH$_4$ plumes are visible from this gas processing facility, one emanating from a source beyond the east edge of the AVIRIS-NG scene This example was associated with a planned maintenance operation, which resulted to a large temporary CH$_4$ plume that was recorded and reported through the normal Greenhouse Gas Reporting Program (Williams, 2016). A second plume is visible at a location shown by the black box in Figure 8a), indicating white pipes associated with an interstate pipeline as the likely emission source (Figure 8c and d).

A H$_2$O retrieval was also performed for this scene and did not reveal enhancements collocated with the CH$_4$ plumes. For all subsequent examples, H$_2$O retrievals were performed but will be shown only in cases where H$_2$O plumes were observed (see Section 5.3). As shown in Figure 8a, the CH$_4$ plumes cross over many land cover types with variable brightness and very dark surfaces resulted in anomalously high retrievals. CH$_4$ results from radiances less than 0.01 $\mu$Wcm$^{-2}$ sr$^{-1}$ nm$^{-1}$ for any band of the CH$_4$ fitting window, corresponding to shadows and water, were removed from the results shown in Figure 8b.

In Figure 9b and e, CH$_4$ emissions from a tank were observed on 19 and 21 April 2015 at 2.8 and 3.2 km AGL (pixel resolutions of 2.6 and 3.0 m respectively). The Google Earth close up shown in Figure 9d indicates a tank as the likely emission source, which was confirmed by the ground crew using a thermal imaging camera on multiple days. Video A1 (see supplement) was acquired on 21 April 2015 at around 18:00 UTC and clearly shows a CH$_4$ plume originating at the top of the tank that is consistent with the AVIRIS-NG CH$_4$ plume observed the same day.

In Frankenberg et al. (2016), CH$_4$ emissions from a pipeline leak were presented (see Figure 4 and Movie S2 in Frankenberg et al. (2016)) and subsequent to publication another suspected pipeline leak (Figure S6 in Frankenberg et al. (2016)) was





confirmed (Karen Spray, Department of Energy, personal communication, 2016). That leak was independently identified and
repaired by the operator as a part of their normal operations prior to publication. In Figure 10b, the $CH_4$ plume from the 19
April 2015 flight at 3.0 km AGL (2.7 m pixel resolution) does not appear associated with visible infrastructure and subsequent
investigation by the ground crews identified the plume origin on 24 April 2015 using the thermal camera (Video A2, see
supplement). This location was along a marked, buried natural gas pipeline and was subsequently confirmed as a pipeline
leak and ultimately shut down for repairs by the local pipeline operators. The estimated flux for this example is 28 kg hr$^{-1}$
Frankenberg et al. (2016) which would result in an estimated annual loss of 13.2 million cubic feet, equivalent to 100,000 USD
(assuming constant annual flux and average cost of 7.40 USD per thousand cubic feet).

### 5.2 Geological CH$_4$ emissions

AVIRIS has been used for quantitative retrievals of $CH_4$ for marine seeps (Roberts et al., 2010; Thorpe et al., 2014) and more
recently a plume observed with AVIRIS-NG was verified as a geological source (see Figure S6 in Frankenberg et al. (2016)).
Subsequent analysis of the Four Corners data set revealed another $CH_4$ plume from a confirmed geological source at Moving
Mountain near Durango, Colorado (LTE, 2015). This AVIRIS-NG scene was acquired at 1.3 km AGL (1 m pixel resolution)
and shows a 10 m long plume (Figure 5b).

### 5.3 CO$_2$ and H$_2$O emissions from power plants

While Dennison et al. (2013) demonstrated the ability of AVIRIS for high resolution mapping of $CO_2$ plumes, in this study
we present two examples using quantitative retrievals. The first example is from the coal-fired San Juan Generating Station
near Farmington, New Mexico that was flown on 20 April 2015 at 1.2 km AGL. Two $CO_2$ plumes are clearly visible in Figure
6b and correspond to two flue-gas stacks that appear active given visible emissions in the true color image (Figure 6a, c). A
third flue-gas stack appears inactive (Figure 6a) with no visible $CO_2$ plume (Figure 6b). The San Juan Generating Station
reported 2015 emissions of 9,843 kt of $CO_2$, equivalent to a flux of 1,123,666 kg $CO_2$ hr$^{-1}$ (EPA, 2016b). An example of a
$CO_2$ retrieval fit and the residual is shown in (Figure 7).

The second example is from a 12 September 2014 flight that included the coal-fired Craig Station near Craig, Colorado. $CO_2$
plumes are visible from flue-gas stacks (Figure 11b) and extend more than 1 km downwind. This power plant reported 2014
emissions of 9,300 kt of $CO_2$, equivalent to a flux of 1,061,644 kg $CO_2$ hr$^{-1}$ (EPA, 2016b). Within the same scene, a $H_2O$ plume
is also visible (Figure 11d) emanating from a number of cooling towers located adjacent to two large cooling ponds (Figure
12a). $CH_4$ retrieval results are also shown in Figure 11c indicating $CH_4$ plumes are not visible in the scene and emphasizing
the ability of these retrievals to distinguish between $CH_4$ and $H_2O$ despite spectral mixing (see Figure 1). Results for dark
surfaces like the cooling ponds were removed from Figure 11b by excluding radiances less than 0.10 $\mu$Wcm$^{-2}$ sr$^{-1}$ nm$^{-1}$ for
any band of the $CO_2$ fitting window, for radiances less than 0.002 $\mu$Wcm$^{-2}$ sr$^{-1}$ nm$^{-1}$ for any band of the $H_2O$ fitting window
(Figure 11d), and for radiances less than 0.01 $\mu$Wcm$^{-2}$ sr$^{-1}$ nm$^{-1}$ for any band of the $CH_4$ fitting window (Figure 11c).

In Figure 12a, the AVIRIS-NG true color image is shown for the close up indicated by the black box in Figure 11. The
flue-gas stacks are visible in the lower left as $CO_2$ sources and cooling towers in the upper right as $H_2O$ sources. Ellipses





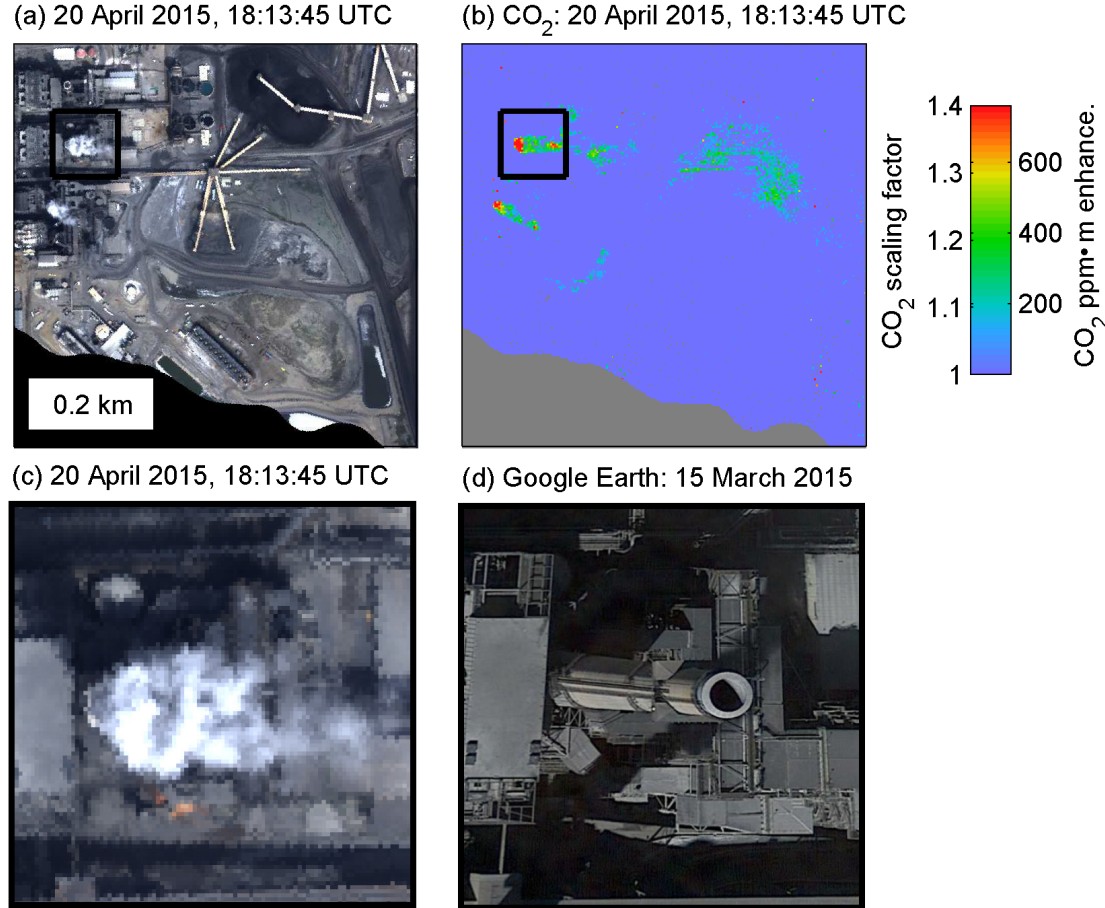

**Figure 6.** (a) AVIRIS-NG true color image subset. (b) $CO_2$ plume is visible. (c) Close up of AVIRIS-NG true color image. (d) Higher resolution Google Earth imagery provides additional spatial context. For all images, north is up.

delineate the shapes of plumes visible in the true color images for the flue-gas stacks (red) and cooling towers (blue). The arrows indicate winds to the southeast for the flue-gas stacks (consistent with $CO_2$ plumes in Figure 11b) and to the east for the cooling towers (consistent with $H_2O$ plumes in Figure 11d). In 12b, the higher resolution Google Earth imagery clearly indicates the flue-gas stacks are much taller (182 m) than the cooling tower (TRI, 2016) based on assessment of shadows, which could explain variable wind directions at the flue-gas stacks and cooling towers.

**6  Conclusions**

In this study, we use the airborne imaging spectrometer AVIRIS-NG and the Iterative Maximum a Posteriori Differential Optical Absorption Spectroscopy (IMAP-DOAS) retrieval to generate gas concentration maps for observed $CH_4$, $CO_2$, and




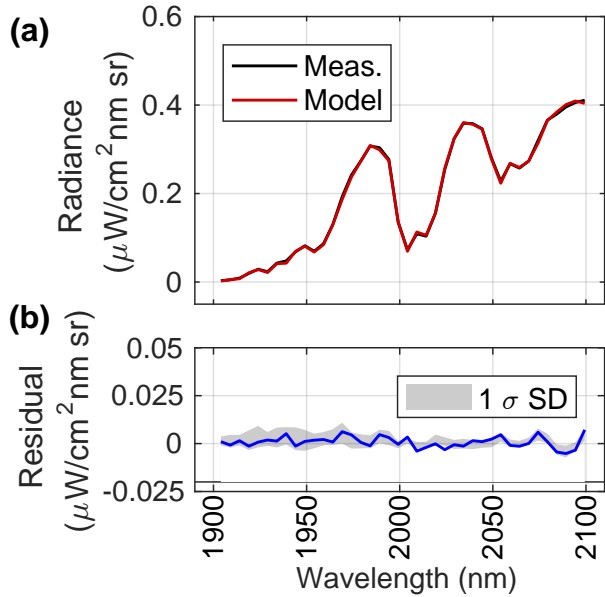

**Figure 7.** (a) AVIRIS-NG measured and modeled radiance for one image pixel within the $CO_2$ plume for the $CO_2$ retrieval (see Figure 6b). (b) The residual is plotted with 1 $\sigma$ standard deviation boundary calculated from residuals for the entire scene.

$H_2O$ plumes. While more than 250 $CH_4$ plumes were observed in the San Juan Basin near Four Corners (Frankenberg et al., 2016), this study focused on a few results from anthropogenic and natural sources, including emissions from mine ventilation

shafts, a gas processing plant, tank, pipeline leak, and natural seep. In addition, $CO_2$ emissions were observed from the flue-stacks of two coal-fired power plants and a $H_2O$ plume was mapped for the cooling towers for one power plant. Observed plumes were consistent with know and suspected emission sources verified by true color AVIRIS-NG imagery and higher resolution Google Earth imagery.

AVIRIS-NG has the high spatial resolution necessary to resolve small-scale emissions and can map large regions quickly, covering the 2,500 km² Four Corners study in approximately two days (9.2 flight hours). This capability is aided by real time detection and geolocation of gas plumes, permitting unambiguous identification of individual emission source locations and communication to ground teams for rapid follow up. This permitted verification of a number of emission sources presented in

5  this study using a thermal camera, including a tank and buried natural gas pipeline. The AVIRIS and AVIRIS-NG instruments have demonstrated $CH_4$ plume mapping capabilities at multiple flight altitudes, ranging from as low as 0.4 km to 3.8 km AGL (0.4 to 3.8 m pixels) for a controlled release experiment (Thorpe et al., 2016a) to 9 km AGL for the Coal Oil Point marine seeps (Thorpe et al., 2014). AVIRIS observed the Aliso Canyon leak on multiple flight days at 6.6 km AGL (6.6 m pixels) while the Hyperion imaging spectrometer, also 10 nm spectral resolution but 30 m pixels, mapped the plume and demonstrated the potential for a space-based application (Thompson et al., 2016).





This study demonstrates a comprehensive greenhouse gas monitoring capability that targets $CH_4$ and $CO_2$, the two dominant anthropogenic climate-forcing agents. The ability to identify individual point source locations of $CH_4$ and $CO_2$ emissions has relevance to the research community as well as the private sector. Understanding the spatial and temporal distribution as well as the magnitude of these emissions is of interest given the large uncertainties associated with anthropogenic emissions. This includes industrial point source emissions of $CH_4$ and $CO_2$, $CH_4$ from oil and gas operations as well as natural gas distribution

and storage, $CH_4$ from agricultural sources, and $CH_4$ and $CO_2$ from landfills. Site operators could identify and mitigate $CH_4$ emissions, which reflect both a potential safety hazard and lost revenue. Water vapor results demonstrate the ability of these retrievals to distinguish between $CH_4$ and $H_2O$ despite spectral mixing in the short wave infrared while offering the potential to improve atmospheric correction and reflectance retrievals with application to the fields of ecology and meteorology.

Despite these promising results, an imaging spectrometer built exclusively for quantitative mapping of gas plumes would

have improved sensitivity compared to AVIRIS-NG (Thorpe et al., 2014). For example, an instrument providing a 1 nm spectral resolution and sampling (2,000-2,400 micron) would permit mapping $CH_4$, $CO_2$, $H_2O$, CO, and $N_2O$ from more diffuse sources using both airborne and orbital platforms (Thorpe et al., 2016b) . The ability to identify emission sources offers the potential to constrain regional greenhouse gas budgets and improve partitioning between anthropogenic and natural emission sources. Because the $CH_4$ lifetime is only about 9 years and $CH_4$ has a high Global Warming Potential, targeting reductions in anthropogenic $CH_4$ emissions offers an effective approach to decrease overall atmospheric radiative forcing.

## 7   Data availability

The AVIRIS-NG data used in this study are available upon request at http:// avirisng.jpl.nasa.gov/ or http://aviris.jpl.nasa.gov/.

## Appendix A:  Appendix A

This appendix contains additional figures referenced in Section  5.

### A1   $CH_4$ emissions from gas processing facility

### A2   $CH_4$ emissions from tank

### A3   $CH_4$ emissions from pipeline leak

### A4   $CO_2$ and $H_2O$ emissions from power plant

*Author contributions.*  C.F., A.K.T. designed research; C.F., A.D.A., A.K.T., D.R.T., B.D.B., R.O.G., E.A.K., C.S., S.C. flight campaign support; A.K.T, C.F., D.R.T., K.G., T.K., J.B. performed research, R.M.D., R.O.G, K.G., T.K., J.B., D.A.R., P.E.D. advised research, A.K.T. and C.F. analyzed data and wrote the paper.





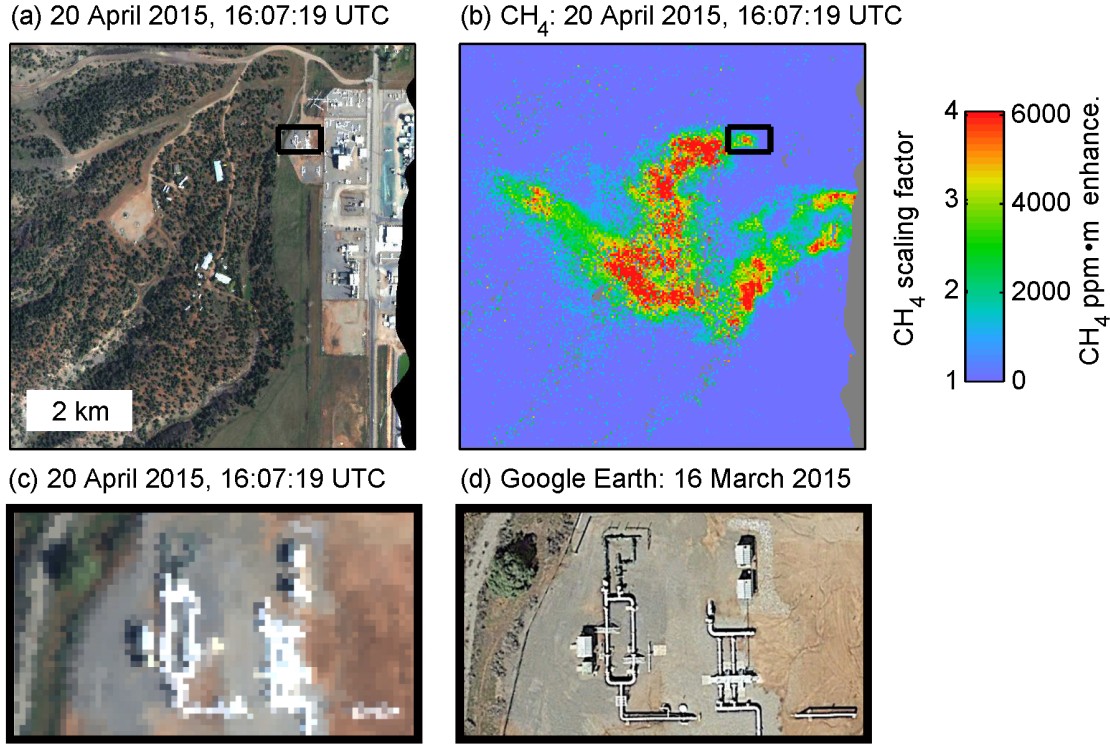

**Figure 8.** (a) AVIRIS-NG true color image subset. (b) Multiple $CH_4$ plumes are visible from this gas processing facility, one emanating from a source beyond the east edge of the AVIRIS-NG scene. A second plume is visible at a location shown by the black box. (c) Close up of AVIRIS-NG true color image indicates white pipes associated with an interstate pipeline as the likely emission source. (d) Higher resolution Google Earth imagery provides additional spatial context. For all images, north is up.

*Competing interests.* The authors declare that they have no conflict of interest.

*Acknowledgements.* The authors thank NASA HQ and Jack Kaye for funding the flight campaign. We would like to acknowledge the contributions of the AVIRIS-NG flight and instrument teams, including Michael Eastwood, Sarah Lundeen, Ian Mccubin, Mark Helmlinger, Scott Nolte, and Betina Pavri. We would also like to thank Simon Hook and Bill Johnson for their support and for the use of the thermal camera. This work was undertaken in part at the Jet Propulsion Laboratory, California Institute of Technology under contract with NASA.
Copyright 2016. All rights reserved.





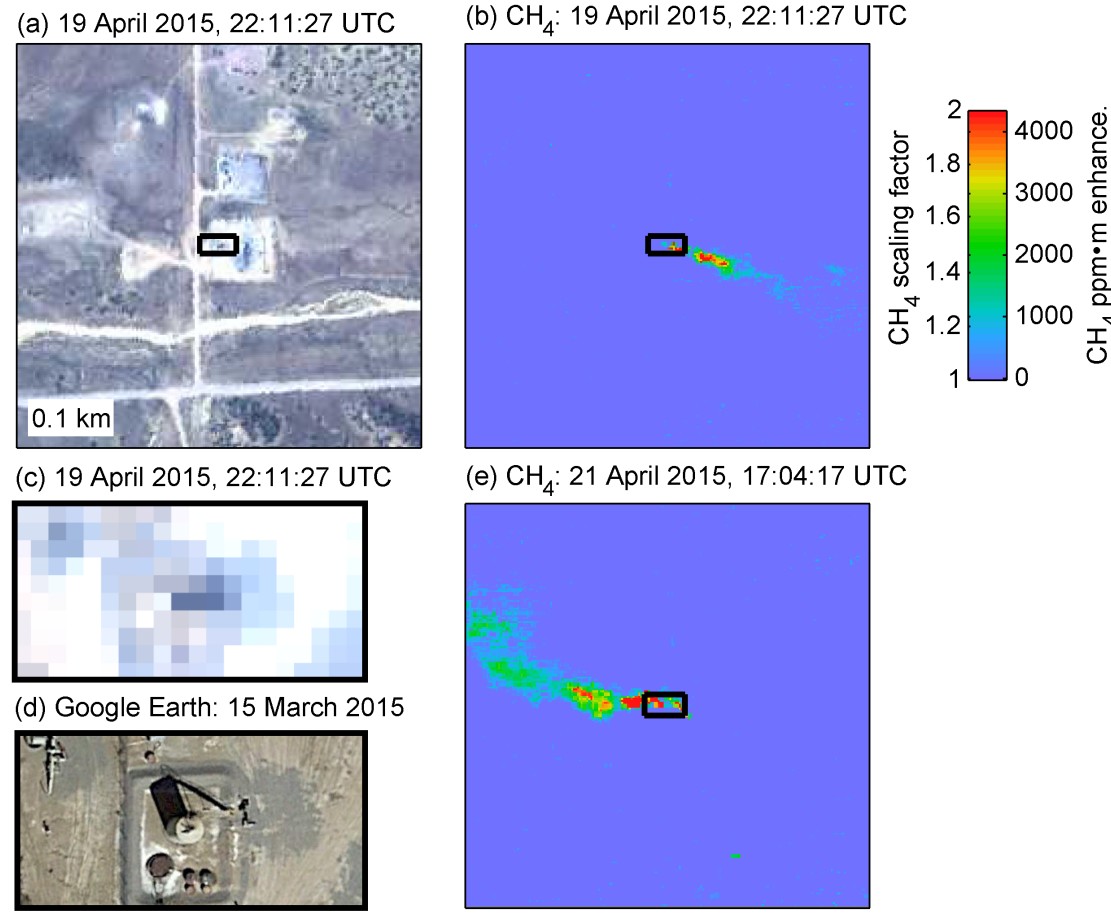

**Figure 9.** (a) 19 April 2015 AVIRIS-NG true color image subset. (b) Prominent $CH_4$ plume visible from location indicated by the black box. (c) Close up of 19 April 2015 AVIRIS-NG true color image. (d) Higher resolution Google Earth imagery indicates the emission source is a tank. (e) 21 April 2015 scene indicates a $CH_4$ plume from the same source with an different orientation due to changes in wind direction. For all images, north is up. A thermal camera video for this source is shown in Video A1.

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



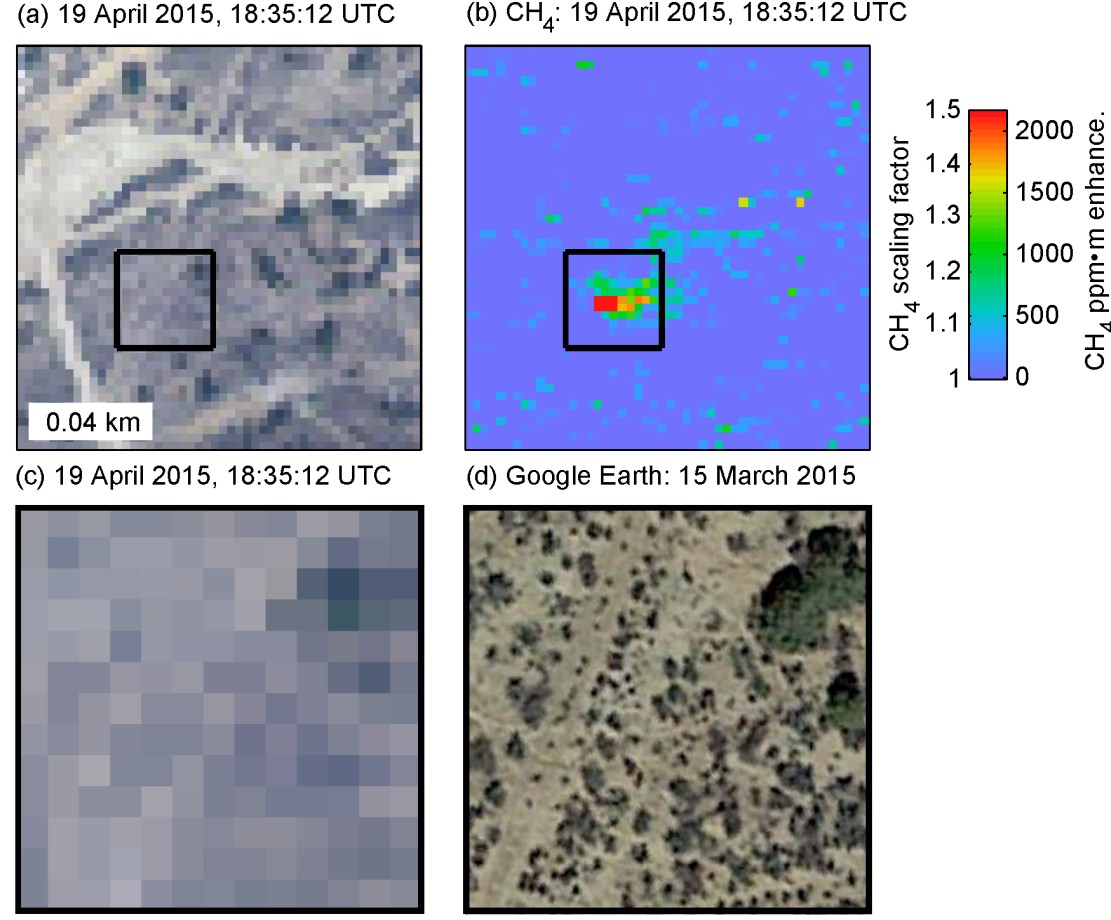

**Figure 10.** (a) AVIRIS-NG true color image subset. (b) A $CH_4$ plume is visible for a confirmed leak from a buried natural gas pipeline. (c) Close up of AVIRIS-NG true color image. (d) Higher resolution Google Earth imagery does not indicate visible infrastucture. For all images, north is up. A thermal camera video for this source is shown in Video A2.

Bovensmann, H., Burrows, J. P., Buchwitz, M., Frerick, J., Noel, S., Rozanov, V. V., Chance, K. V., and Goede, A. P. H.: SCIA-MACHY: Mission Objectives and Measurement Modes, Journal of the Atmospheric Sciences, 56, 127–150, doi:10.1175/1520-0469(1999)056<0127:SMOAMM>2.0.CO;2, 1999.

Brandt, A. R., Heath, G. A., Kort, E. A., O'Sullivan, F., Pétron, G., Jordaan, S. M., Tans, P., Wilcox, J., Gopstein, A. M., Arent, D., Wofsy, S., Brown, N. J., Bradley, R., Stucky, G. D., Eardley, D., and Harriss, R.: Methane leaks from North American natural gas systems, Science, 343, 733–735, doi:10.1126/science.1247045, 2014.

Brandt, A. R., Heath, G. A., and Cooley, D.: Methane leaks from natural gas systems follow extreme distributions, Environmental Science & Technology, 50, 12 512–12 520, doi:10.1021/acs.est.6b04303, 2016.



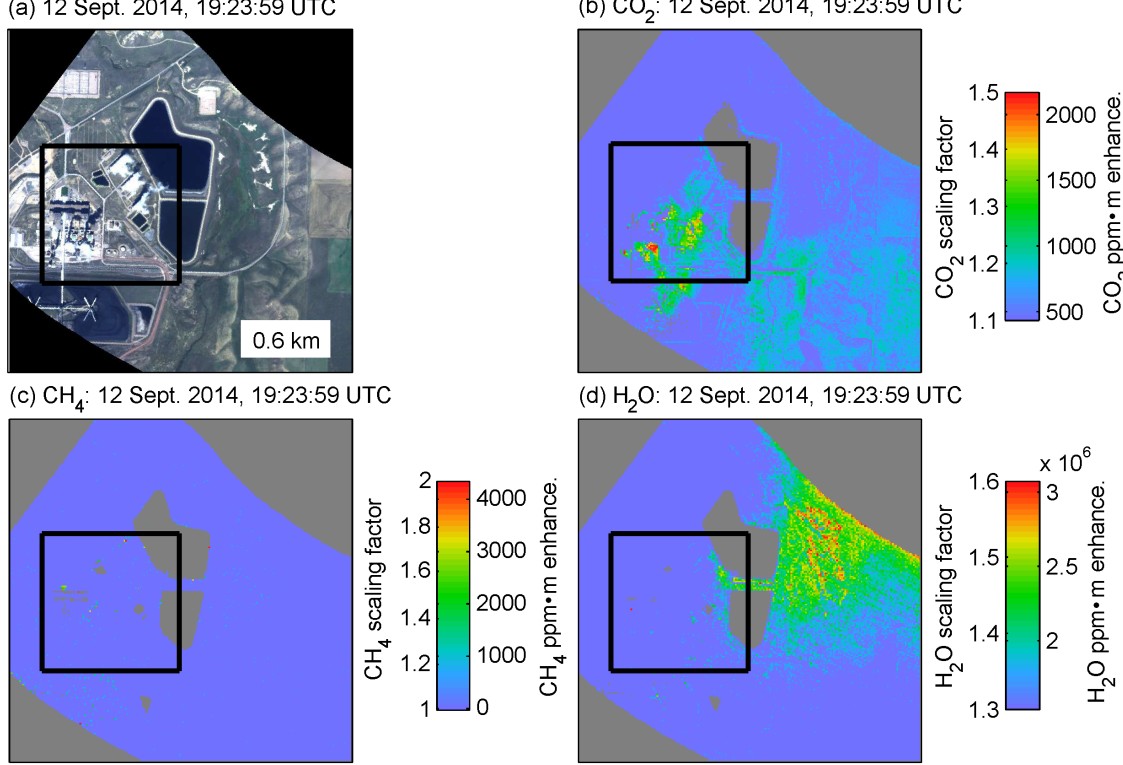

**Figure 11.** (a) AVIRIS-NG true color image subset. (b) $CO_2$ plumes are visible emanating from flue-gas stacks. (c) $CH_4$ retrieval results. (d) $H_2O$ plume visible from cooling towers (see Figure 12). For all images, north is up.

Caulton, D. R., Shepson, P. B., Santoro, R. L., Sparks, J. P., Howarth, R. W., Ingraffea, A. R., Cambaliza, M. O. L., Sweeney, C., Karion, A., Davis, K. J., Stirm, B. H., Montzka, S. A., and Miller, B. R.: Toward a better understanding and quantification of methane emissions from shale gas development, Proceedings of the National Academy of Sciences, doi:10.1073/pnas.1316546111, 2014.

Ciais, P., Crisp, D., van der Gron, H. D., Engelen, R., Janssens-Maenhout, G., Heimann, M., Rayner, P., , and Scholze, M.: Towards a European operational observing system to monitor fossil CO2 emissions, Final report from the expert group, European Comission, Tech. rep., European Comission, 2015.

Conley, S., Franco, G., Faloona, I., Blake, D. R., Peischl, J., and Ryerson, T. B.: Methane emissions from the 2015 Aliso Canyon blowout in
Los Angeles, CA, Science, 351, 1317–1320, doi:10.1126/science.aaf2348, 2016.

Dennison, P. E., Thorpe, A. K., Pardyjak, E. R., Roberts, D. A., Qi, Y., Green, R. O., Bradley, E. S., and Funk, C. C.: High spatial resolution mapping of elevated atmospheric carbon dioxide using airborne imaging spectroscopy: Radiative transfer modeling and power plant plume detection, Remote Sensing of Environment, 139, 116–129, doi:10.1016/j.rse.2013.08.001, 2013.

Deschamps, A., Marion, R., Briottet, X., Foucher, P., and Lavigne, C.: Simultaneous CO2 and aerosol retrieval in a vegetation fire plume
using AVIRIS hyperspectral data, doi:10.1109/WHISPERS.2011.6080896, 2011.

EIA: U.S. Energy Information Administraton, EIA annual energy outlook 2013, Tech. rep., U.S. Energy Information Administraton, 2013.

EIA: U.S. Energy Information Administraton, Top 100 U.S. oil and gas fields, Tech. rep., U.S. Energy Information Administraton, 2015.

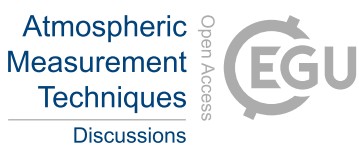

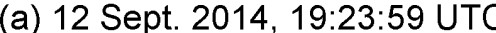

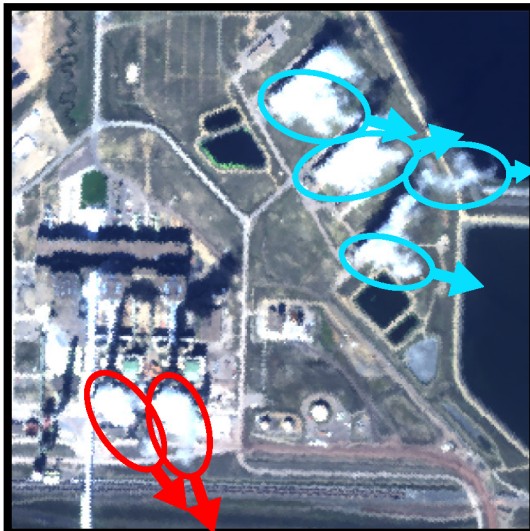 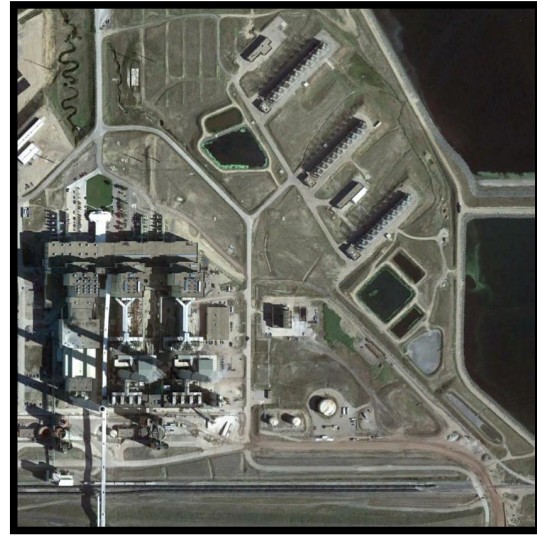

**Figure 12.** (a) AVIRIS-NG true color image for close up indicated by black box in Figure 11. Flue-gas stacks visible in lower left as $CO_2$ sources and cooling towers in upper right as $H_2O$ sources. Ellipses delineate shapes of plumes visible in true color images for the flue-gas stacks (red) and cooling towers (blue). The arrows indicate winds to the southeast for the flue-gas stacks (consistent with $CO_2$ plumes in Figure 11b) and to the east for the cooling towers (consistent with $H_2O$ plumes in Figure 11d). (b) Higher resolution Google Earth imagery clearly indicates the flue-gas stacks are much taller than the cooling towers based on assessment of shadows. For both images, north is up.

EPA: U.S. Environmental Protection Agency, Inventory of U.S. Greenhouse Gas Emissions and Sinks: 1990–2014, Technical Report EPA 430-R-16-002 (Environmental Protection Agency), Tech. rep., U.S. Environmental Protection Agency, 2016a.

EPA: U.S. Environmental Protection Agency, 2015 Greenhouse Gas Emissions from Large Facilities, https://ghgdata.epa.gov/, Tech. rep., U.S. Environmental Protection Agency, 2016b.

Frankenberg, C., Platt, U., and Wagner, T.: Iterative maximum a posteriori (IMAP)-DOAS for retrieval of strongly absorbing trace gases: Model studies for CH4 and CO2 retrieval from near infrared spectra of SCIAMACHY onboard ENVISAT, Atmospheric Chemistry and Physics, 5, 9–22, doi:10.5194/acp-5-9-2005, 2005.

Frankenberg, C., Thorpe, A. K., Thompson, D. R., Hulley, G., Kort, E. A., Vance, N., Borchardt, J., Krings, T., Gerilowski, K., Sweeney, C., Conley, S., Bue, B. D., Aubrey, A. D., Hook, S., and Green, R. O.: Airborne methane remote measurements reveal heavy-tail flux distribution in Four Corners region, Proceedings of the National Academy of Sciences of the United States of America, 113, 9734–9739, doi:10.1073/pnas.1605617113, 2016.

Galfalk, M., Olofsson, G., Crill, P., and Bastviken, D.: Making methane visible, Nature Climate Change, 6, 426–430, 25   doi:10.1038/nclimate2877, 2016.

Gao, B. C. and Goetz, A. F. H.: Column atmospheric water-vapor and vegetation liquid water retrievals from airborne imaging spectrometer data, Journal of Geophysical Research-Atmospheres, 95, 3549–3564, doi:10.1029/JD095iD04p03549, 1990.





Gerilowski, K., Tretner, A., Krings, T., Buchwitz, M., Bertagnolio, P. P., Belemezov, F., Erzinger, J., Burrows, J. P., and Bovensmann, H.: MAMAP - a new spectrometer system for column-averaged methane and carbon dioxide observations from aircraft: instrument description and performance analysis, Atmospheric Measurement Techniques, 4, 215–243, doi:10.5194/amt-4-215-2011, 2011.

Green, R. O., Eastwood, M. L., Sarture, C. M., Chrien, T. G., Aronsson, M., Chippendale, B. J., Faust, J. A., Pavri, B. E., Chovit, C. J., Solis, M. S., Olah, M. R., and Williams, O.: Imaging spectroscopy and the Airborne Visible Infrared Imaging Spectrometer (AVIRIS), Remote Sensing of Environment, 65, 227–248, doi:10.1016/S0034-4257(98)00064-9, 1998.

Hamlin, L., Green, R., Mouroulis, P., Eastwood, M., Wilson, D., Dudik, M., and Paine, C.: Imaging spectrometer science measurements for terrestrial ecology: AVIRIS and new developments, doi:10.1109/AERO.2011.5747395, 2011.

Hopkins, F. M., Kort, E. A., Bush, S. E., Ehleringer, J. R., Lai, C. T., Blake, D. R., and Randerson, J. T.: Spatial patterns and source attribution of urban methane in the Los Angeles Basin, Journal of Geophysical Research-Atmospheres, 121, 2490–2507, doi:10.1002/2015JD024429, 2016.

Hulley, G. C., Duren, R. M., Hopkins, F. M., Hook, S. J., Vance, N., Guillevic, P., Johnson, W. R., Eng, B. T., Mihaly, J. M., Jovanovic, V. M., Chazanoff, S. L., Staniszewski, Z. K., Kuai, L., Worden, J., Frankenberg, C., Rivera, G., Aubrey, A. D., Miller, C. E., Malakar, N. K., Tomas, J. M. S., and Holmes, K. T.: High spatial resolution imaging of methane and other trace gases with the airborne Hyperspectral Thermal Emission Spectrometer (HyTES), Atmospheric Measurement Techniques, 9, 2393–2408, doi:10.5194/amt-9-2393-2016, 2016.

Jackson, R. B., Down, A., Phillips, N. G., Ackley, R. C., Cook, C. W., Plata, D. L., and Zhao, K. G.: Natural gas pipeline leaks across Washington, DC, Environmental Science and Technology, 48, 2051–2058, doi:10.1021/es404474x, 2014.

Johnson, D. R., Covington, A. N., and Clark, N. N.: Methane Emissions from leak and loss audits of natural gas compressor stations and storage facilities, Environmental Science and Technology, 49, 8132–8138, doi:10.1021/es506163m, 2015.

Kalnay, E., Kanamitsu, M., Kistler, R., Collins, W., Deaven, D., Gandin, L., Iredell, M., Saha, S., White, G., Woollen, J., Zhu, Y., Chelliah, M., Ebisuzaki, W., Higgins, W., Janowiak, J., Mo, K. C., Ropelewski, C., Wang, J., Leetmaa, A., Reynolds, R., Jenne, R., and Joseph, D.: The NCEP/NCAR 40-year reanalysis project, Bulletin of the American Meteorological Society, 77, 437–471, doi:10.1175/1520-0477(1996)077<0437:TNYRP>2.0.CO;2, 1996.

Karion, A., Sweeney, C., Petron, G., Frost, G., Hardesty, R. M., Kofler, J., Miller, B. R., Newberger, T., Wolter, S., Banta, R., Brewer, A., Dlugokencky, E., Lang, P., Montzka, S. A., Schnell, R., Tans, P., Trainer, M., Zamora, R., and Conley, S.: Methane emissions estimate from airborne measurements over a western United States natural gas field, Geophysical Research Letters, 40, 4393–4397, doi:10.1002/grl.50811, 2013.

Kirschke, S., Bousquet, P., Ciais, P., Saunois, M., Canadell, J. G., Dlugokencky, E. J., Bergamaschi, P., Bergmann, D., Blake, D. R., Bruhwiler, L., Cameron-Smith, P., Castaldi, S., Chevallier, F., Feng, L., Fraser, A., Heimann, M., Hodson, E. L., Houweling, S., Josse, B., Fraser, P. J., Krummel, P. B., Lamarque, J. F., Langenfelds, R. L., Le Quere, C., Naik, V., O'Doherty, S., Palmer, P. I., Pison, I., Plummer, D., Poulter, B., Prinn, R. G., Rigby, M., Ringeval, B., Santini, M., Schmidt, M., Shindell, D. T., Simpson, I. J., Spahni, R., Steele, L. P., Strode, S. A., Sudo, K., Szopa, S., van der Werf, G. R., Voulgarakis, A., van Weele, M., Weiss, R. F., Williams, J. E., and Zeng, G.: Three decades of global methane sources and sinks, Nature Geoscience, 6, 813–823, doi:10.1038/ngeo1955, 2013.

Kneizys, F. X., Abreu, L. W., Anderson, G. P., Chetwynd, J. H., Shettle, E. P., Robertson, D. C., Acharya, P., Rothman, L., Selby, J. E. A., Gallery, W. O., and Clough, S. A.: The MODTRAN 2/3 report and LOWTRAN 7 model, Tech. rep., Tech. rep., Phillips Laboratory, Geophysics Directorate, 1996.

Kort, E. A., Frankenberg, C., Costigan, K. R., Lindenmaier, R., Dubey, M. K., and Wunch, D.: Four Corners: The largest US methane anomaly viewed from space, Geophysical Research Letters, p. 2014GL061503, doi:10.1002/2014GL061503, 2014.



Krautwurst, S., Gerilowski, K., Jonsson, H. H., Thompson, D. R., Kolyer, R. W., Thorpe, A. K., Horstjann, M., Eastwood, M., Leifer, I., Vigil, S., Krings, T., Borchardt, J., Buchwitz, M., Fladeland, M. M., Burrows, J. P., and Bovensmann, H.: Methane emissions from a Cal-
ifornia landfill, determined from airborne remote sensing and in-situ measurements, Atmospheric Measurement Techniques Discussions, doi:10.5194/amt-2016-391, 2016.

Krings, T., Gerilowski, K., Buchwitz, M., Reuter, M., Tretner, A., Erzinger, J., Heinze, D., Pfluger, U., Burrows, J. P., and Bovensmann, H.: MAMAP – a new spectrometer system for column-averaged methane and carbon dioxide observations from aircraft: retrieval algorithm and first inversions for point source emission rates, Atmospheric Measurement Techniques, 4, 1735–1758, doi:10.5194/amt-4-1735-2011,
35      2011.

Krings, T., Gerilowski, K., Buchwitz, M., Hartmann, J., Sachs, T., Erzinger, J., Burrows, J. P., and Bovensmann, H.: Quantification of methane emission rates from coal mine ventilation shafts using airborne remote sensing data, Atmospheric Measurement Techniques, 6, 151–166, doi:10.5194/amt-6-151-2013, 2013.

Kuai, L., Worden, J. R., Li, K. F., Hulley, G. C., Hopkins, F. M., Miller, C. E., Hook, S. J., Duren, R. M., and Aubrey, A. D.: Characterization of anthropogenic methane plumes with the Hyperspectral Thermal Emission Spectrometer (HyTES): A retrieval method and error analysis, Atmospheric Measurement Techniques, 9, 3165–3173, doi:10.5194/amt-9-3165-2016, 2016.

Lavoie, T. N., Shepson, P. B., Cambaliza, M. O. L., Stirm, B. H., Karion, A., Sweeney, C., Yacovitch, T. I., Herndon, S. C., Lan, X., and Lyon,
D.: Aircraft-based measurements of point source methane emissions in the Barnett Shale Basin, Environmental Science & Technology, 49, 7904–7913, doi:10.1021/acs.est.5b00410, 2015.

LTE: 2015 Fruitland outcrop monitoring report ,La Plata County, Colorado, Tech. rep., LT Environmental, Inc., http://cogcc.state.co.us/ documents/library/AreaReports/SanJuanBasin/3m_project/2015%20FRUITLAND%20OUTCROP%20MONITORING%20REPORT_ La_Plata.pdf, 2015.

Lyon, D. R., Zavala-Araiza, D., Alvarez, R. A., Harriss, R., Palacios, V., Lan, X., Talbot, R., Lavoie, T., Shepson, P., Yacovitch, T. I., Herndon, S. C., Marchese, A. J., Zimmerle, D., Robinson, A. L., and Hamburg, S. P.: Constructing a Spatially Resolved Methane Emission Inventory for the Barnett Shale Region, Environmental Science & Technology, 49, 8147–8157, doi:10.1021/es506359c, 2015.

Marion, R., Michel, W., and Faye, C.: Measuring trace gases in plumes from hyperspectral remotely sensed data, Ieee Transactions on Geoscience and Remote Sensing, 42, 854–864, doi:10.1109/TGRS.2003.820604, 2004.

Miller, S., Wofsy, S., Michalak, A., Kort, E. A., Andrews, A., Biraud, S., Dlugockenky, E. J., Eluszkiewicz, J., Fisher, M., Janssens-Maenhout, G., Miller, B., Miller, J., Montzka, S., Nehrkorn, T., and Sweeney, C.: Anthropogenic emissions of methane in the United States, Proceedings of the National Academy of Sciences of the United States of America, 110, doi:10.1073/pnas.1314392110, 2013.

Myhre, G., Shindell, D., Bréon, F.-M., Collins, W., Fuglestvedt, J., Huang, J., Koch, D., Lamarque, J., Lee, D., Mendoza, B., Nakajima, T., Robock, A., Stephens, G., Takemura, T., and Zhang, H.: Anthropogenic and natural radiative forcing. In: Climate change 2013: The
physical science basis. Contribution of Working Group I to the Fifth Assessment Report of the Intergovernmental Panel on Climate Change, Tech. rep., Intergovernmental Panel on Climate Change, 2013.

Nisbet, E. G., Dlugokencky, E. J., and Bousquet, P.: Methane on the rise—again, Science, 343, 493–495, doi:10.1126/science.1247828, 2014.

NOAA: GMD measurement locations, National Oceanic & Atmospheric Administration (NOAA), Earth System Research Laboratory, Global Monitoring Division, Tech. rep., National Oceanic & Atmospheric Administration, 2015.

NRC: National Research Council committee on methods for estimating greenhouse gas emissions; Verifying greenhouse gas emissions: Methods to support international climate agreements, The National Academies Press, Washington, D.C., 2010.



Ogunjemiyo, S., Roberts, D. A., Keightley, K., Ustin, S. L., Hinckley, T., and Lamb, B.: Evaluating the relationship between AVIRIS water vapor and poplar plantation evapotranspiration, Journal of Geophysical Research, 107, doi:10.1029/2001JD001194, 2004.

Phillips, N. G., Ackley, R., Crosson, E. R., Down, A., Hutyra, L. R., Brondfield, M., Karr, J. D., Zhao, K. G., and Jackson, R. B.: Mapping
urban pipeline leaks: Methane leaks across Boston, Environmental Pollution, 173, 1–4, doi:10.1016/j.envpol.2012.11.003, 2013.

Rella, C. W., Tsai, T. R., Botkin, C. G., Crosson, E. R., and Steele, D.: Measuring emissions from oil and natural gas well pads using the mobile flux plane technique, Environmental Science & Technology, 49, 4742–4748, doi:10.1021/acs.est.5b00099, 2015.

Roberts, D. A., Bradley, E. S., Cheung, R., Leifer, I., Dennison, P. E., and Margolis, J. S.: Mapping methane emissions from a marine geological seep source using imaging spectrometry, Remote Sensing of Environment, 114, 592–606, doi:10.1016/j.rse.2009.10.015, 2010.

Rodgers, C. D.: Inverse methods for atmospheric sounding, theory and practice, World Scientific, London, 2000.

Rothman, L. S., Gordon, I. E., Barbe, A., Benner, D. C., Bernath, P. E., Birk, M., Boudon, V., Brown, L. R., Campargue, A., Champion, J. P., Chance, K., Coudert, L. H., Dana, V., Devi, V. M., Fally, S., Flaud, J. M., Gamache, R. R., Goldman, A., Jacquemart, D., Kleiner, I., Lacome, N., Lafferty, W. J., Mandin, J. Y., Massie, S. T., Mikhailenko, S. N., Miller, C. E., Moazzen-Ahmadi, N., Naumenko, O. V., Nikitin, A. V., Orphal, J., Perevalov, V. I., Perrin, A., Predoi-Cross, A., Rinsland, C. P., Rotger, M., Simeckova, M., Smith, M. A. H., Sung, K., Tashkun, S. A., Tennyson, J., Toth, R. A., Vandaele, A. C., and Vander Auwera, J.: The HITRAN 2008 molecular spectroscopic database, Journal of Quantitative Spectroscopy & Radiative Transfer, 110, 533–572, doi:10.1016/j.jqsrt.2009.02.013, 2009.

Schaefer, H., Mikaloff Fletcher, S. E., Veidt, C., Lassey, K. R., Brailsford, G. W., Bromley, T. M., Dlugokencky, E. J., Michel, S. E., Miller,
J. B., Levin, I., Lowe, D. C., Martin, R. J., Vaughn, B. H., and White, J. W. C.: A 21st century shift from fossil-fuel to biogenic methane emissions indicated by 13CH4, Science, doi:10.1126/science.aad2705, 2016.

Schwietzke, S., Sherwood, O. A., Bruhwiler, L. M. P., Miller, J. B., Etiope, G., Dlugokencky, E. J., Michel, S. E., Arling, V. A., Vaughn, B. H., White, J. W. C., and Tans, P. P.: Upward revision of global fossil fuel methane emissions based on isotope database, Nature, 538, doi:10.1038/nature19797, 2016.

Smith, M. L., Kort, E. A., Karion, A., Sweeney, C., Herndon, S. C., and Yacovitch, T. I.: Airborne ethane observations in the Barnett Shale: Quantification of ethane flux and attribution of methane emissions, Environmental Science & Technology, 49, 8158–8166, doi:10.1021/acs.est.5b00219, 2015.

Thompson, D. R., Gao, B. C., Green, R. O., Dennison, P. E., Roberts, D. A., and Lundeen, S.: Atmospheric Correction for Global Mapping Spectroscopy: Advances for the HyspIRI Preparatory Campaign, Remote Sensing of Environment, 167, 64–77,
doi:10.1016/j.rse.2015.02.010, 2015a.

Thompson, D. R., Leifer, I., Bovensmann, H., Eastwood, M., Fladeland, M., Frankenberg, C., Gerilowski, K., Green, R. O., Kratwurst, S., Krings, T., Luna, B., and Thorpe, A. K.: Real-time remote detection and measurement for airborne imaging spectroscopy: a case study with methane, Atmospheric Measurement Techniques, 8, 4383–4397, doi:10.5194/amt-8-4383-2015, 2015b.

Thompson, D. R., Thorpe, A. K., Frankenberg, C., Green, R. O., Duren, R., Guanter, L., Hollstein, A., Middleton, E., Ong, L., and Ungar,
S.: Space-based remote imaging spectroscopy of the Aliso Canyon CH4 superemitter, Geophysical Research Letters, 43, 6571–6578, doi:10.1002/2016GL069079, 2016.

Thorpe, A., Frankenberg, C., and Roberts, D.: Retrieval techniques for airborne imaging of methane concentrations using high spatial and moderate spectral resolution: Application to AVIRIS, Atmospheric Measurement Techniques, 7, 491–506, doi:10.5194/amt-7-491-2014, 2014.

Thorpe, A. K., Frankenberg, C., Aubrey, A. D., Roberts, D. A., Nottrott, A. A., Rahn, T. A., Sauer, J. A., Dubey, M. K., Costigan, K. R., Arata, C., Steffke, A. M., Hills, S., Haselwimmer, C., Charlesworth, D., Funk, C. C., Green, R. O., Lundeen, S. R., Boardman, J. W.,





Eastwood, M. L., Sarture, C. M., Nolte, S. H., Mccubbin, I. B., Thompson, D. R., and McFadden, J. P.: Mapping methane concentrations from a controlled release experiment using the next generation airborne visible/infrared imaging spectrometer (AVIRIS-NG), Remote Sensing of Environment, 179, 104–115, doi:10.1016/j.rse.2016.03.032, 2016a.

Thorpe, A. K., Frankenberg, C., Green, R. O., Thompson, D. R., Aubrey, A. D., Mouroulis, P., Eastwood, M. L., and Matheou, G.: The Airborne Methane Plume Spectrometer (AMPS): Quantitative imaging of methane plumes in real time, 2016 IEEE Aerospace Conference, doi:10.1109/AERO.2016.7500756, 2016b.

Tratt, D. M., Buckland, K. N., Hall, J. L., Johnson, P. D., Keim, E. R., Leifer, I., Westberg, K., and Young, S. J.: Airborne visualization and quantification of discrete methane sources in the environment, Remote Sensing of Environment, 154, 74–88,

doi:10.1016/j.rse.2014.08.011, 2014.

TRI: Tri-State Generation and Transmission Association, http://www.tsgt.coop/AboutUs/baseload-resources.cfm, Tech. rep., Tri-State Generation and Transmission Association, 2016.

Turner, A. J., Jacob, D. J., Wecht, K. J., Maasakkers, J. D., Lundgren, E., Andrews, A. E., Biraud, S. C., Boesch, H., Bowman, K. W., Deutscher, N. M., Dubey, M. K., Griffith, D. W. T., Hase, F., Kuze, A., Notholt, J., Ohyama, H., Parker, R., Payne, V. H., Sussmann, R., Sweeney, C., Velazco, V. A., Warneke, T., Wennberg, P. O., and Wunch, D.: Estimating global and North American methane emissions with high spatial resolution using GOSAT satellite data, Atmospheric Chemistry and Physics, 15, 7049–7069, doi:10.5194/acp-15-7049-2015,

5  2015.

Wecht, K. J., Jacob, D. J., Frankenberg, C., Jiang, Z., and Blake, D. R.: Mapping of North American methane emissions with high spatial resolution by inversion of SCIAMACHY satellite data, Journal of Geophysical Research-Atmospheres, 119, 7741–7756, doi:10.1002/2014JD021551, 2014.

Williams: NASA study shows one detection of methane in multiple surveys of Ignacio plant, Tech. rep., Williams Partners L.P., https:

10  //blog.williams.com/projects-and-operations/nasa-study-shows-no-detection-of-methane-in-multiple-surveys-of-ignacio-plant/, 2016.

Zavala-Araiza, D., Lyon, D., Alvarez, R. A., Palacios, V., Harriss, R., Lan, X., Talbot, R., and Hamburg, S. P.: Toward a functional definition of methane super-emitters: Application to natural gas production sites, Environmental Science & Technology, 49, 8167–8174, doi:10.1021/acs.est.5b00133, 2015.