# Peer review of "Airborne DOAS retrievals of methane, carbon dioxide, and water vapor concentrations at high spatial resolution: application to AVIRIS-NG"

_Atmospheric Measurement Techniques, 2017_

## Referee Comment (RC1) · Anonymous Referee #1 · 31 May 2017

In their paper on airborne DOAS retrievals Andrew K. Thorpe et al. demonstrate that small-scale plumes of greenhouse gases can be detected with their measurement system. Based on established retrieval theory they present an application where the mutual interference of spectral signatures is addressed by jointly fitting the concentrations of the related species. They demonstrate the capability of their method in a predictivist sense, i.e. they compare their retrieved data to use-novel data (data which have not been used in the retrieval as e.g. a priori information) to confirm their results. This approach is valid and apt to furnish evidence that the method chosen is indeed adequate and has the claimed merits. Further, the paper is well written and clearly structured,

and there seems not much to be criticized.

However, I think, the paper still could be improved. Modern retrieval theory offers a lot of diagnostics which would be interesting in the given context: averaging kernels (spatial resolution and content of prior information in the data), retrieval covariance matrices (particularly retrieval error bars and detection limits inferred from these), etc. A thorough discussion of these quantities would probably shift the focus of the paper. Since a reviewer should not dictate the authors the focus of their paper, and since their way to demonstrate the capability of their measurement system and retrieval method is already convincing, I am reluctant to force the authors to include the discussion of these quantitative diagnostic data and only encourage them to consider this issue if they can easily accommodate it. At least typical retrieval errors should be easy to include.

In summary, I recommend publication of this paper in AMT after correction of the following technical issues:

p7 l8: I there a version number of this particular HITRAN data set available?

p8 l2: ...these Jacobians are not shown. ("in Figure 1" can be deleted because it is clear from the context).

p8 l18/19 (attention: line numbering is not monotonic in my version here and on the following pages!) It is reported which gases are "included" in which retrieval. This is, however, ambiguous. This can mean "included in the forward calculation of the retrievals, using their respective a priori profiles and leaving these as they are" or it can mean "included as additional unknown variables of the retrieval". I guess you mean the latter but you should be more specific here.

p10 l7 "An H2O retrieval..."; (replace "A" by "An")

p11 l14 "spectral mixing": Is this an established technical term in your community? In my community the superposition of spectral signatures from various species is typically
called "spectral interference" or we use phrases like "signal from interfering species".

p13 l32 knowN

p14 l6 Appendix A is printed twice.

---

## Referee Comment (RC2) · Anonymous Referee #2 · 26 Jun 2017

**General comments**

In the paper, the authors apply IMAP-DOAS algorithm to the AVIRIS-NG instrument for mapping $CO_2$, $CH_4$ and $H_2O$ plumes for seven sources. The topic of the paper is well within the scope of AMT. The methods are valid and the authors present new data. However, some results are not sufficient to support some interpretation and conclusions in the paper.

The paper lacks a description and discussion of measurement uncertainties making it very difficult to evaluate the quality of the results. The authors repeatedly state that they identified over 250 $CH_4$ plumes with AVIRIS-NG, but they only show four examples where IMAP-DOAS was applied. They authors need to explain the reasons for choosing these four plumes and for omitting the others. Furthermore, the advantage of IMAP-DOAS over the matched filter is not clearly stated giving the impression that IMAP-DOAS is actually inferior to the filter because only four plumes are detectable. To conclude, the authors should describe measurement uncertainties and discuss how IMAP-DOAS compares to and differs from the linearized matched filter approach.

The authors claim that they are able to detect $H_2O$ from cooling towers (Figure A4d). However, I am not able to identify these plumes in Figure A4d. The figure shows a large area of enhanced scaling factors east of the cooling ponds, but this $H_2O$ signal more likely originates from the ponds. I agree that wind directions at cooling towers and stacks can be different due to temporal variability or the height dependency of the wind direction (Ekman spiral). However, since the plumes a very short and likely meandering, it is very difficult to estimate the wind direction from the true color composite alone in Figure A5a. Nonetheless, I think the wind direction at the cooling tower is more southerly than indicated by the blue arrows. To conclude, the authors need to add more support for their claim that $H_2O$ has been detected from the cooling towers.

In summary, I recommend publication of this paper in AMT after adding the points above as well as the following specific comments and technical corrections.

**Specific comments**

- Adding a map showing all measurements sites to the paper would make it easier to locate the sites.

- The authors should change their units from ppm m to a more common unit such

as dry air column averaged mole fractions (XCO2 and XCH4).

**Technical corrections**

- Page 6, Section 5.1, 3rd paragraph: Add full stop after "east edge of the AVIRIS-NG scene"

- Page 7, line 39: Frankenberg et al. (2016) -> (Frankenberg et al. 2016)

---

## Author Comment (AC1) · 29 Aug 2017

We thank the reviewer for their thorough and helpful comments. A detailed response for each comment is provided in the attached document and changes were made to the manuscript. As a result, the manuscript has been significantly strengthened and we very much appreciate your consideration for publication in Atmospheric Measurement Techniques.

Sincerely, Andrew Thorpe and coauthors

[Figure]

Please also note the supplement to this comment:
https://www.atmos-meas-tech-discuss.net/amt-2017-51/amt-2017-51-AC1-supplement.pdf

---

## Author Comment (AC2) · 29 Aug 2017

**Response to reviewer comments**

Anonymous Referee #2 comments:

General comments

In the paper, the authors apply IMAP-DOAS algorithm to the AVIRIS-NG instrument for mapping CO2, CH4 and H2O plumes for seven sources. The topic of the paper is well within the scope of AMT. The methods are valid and the authors present new data. However, some results are not sufficient to support some interpretation and conclusions in the paper. The paper lacks a description and discussion of measurement uncertainties making it very difficult to evaluate the quality of the results.

The authors repeatedly state that they identified over 250 CH4 plumes with AVIRIS-NG, but they only show four examples where IMAP-DOAS was applied. They authors need to explain the reasons for choosing these four plumes and for omitting the others. Furthermore, the advantage of IMAP-DOAS over the matched filter is not clearly stated giving the impression that IMAP-DOAS is actually inferior to the filter because only four plumes are detectable.

**Response:** Thank you for this observation, we can understand that this could generate some confusion. We have added additional text to explain the linearized matched filter, how it differs from the IMAP-DOAS approach, and the rationale for why IMAP-DOAS results were generated for a subset of the plumes identified in a previous study (Frankenberg et al., 2016). Please refer to the first two paragraphs of section 5.1 (page 11).

To conclude, the authors should describe measurement uncertainties and discuss how IMAP-DOAS compares to and differs from the linearized matched filter approach.

**Response:** Reviewer 2 brings up this point which was also discussed by Reviewer 1. A detailed uncertainty analysis between IMAP-DOAS and the linearized matched filter is beyond the scope of this paper. However, a discussion of typical IMAP-DOAS retrieval errors is appropriate and was included (see page 10, line 16 to page 11, line 24).

The authors claim that they are able to detect H2O from cooling towers (Figure A4d). However, I am not able to identify these plumes in Figure A4d. The figure shows a large area of enhanced scaling factors east of the cooling ponds, but this H2O signal more likely originates from the ponds. I agree that wind directions at cooling towers and stacks can be different due to temporal variability or the height dependency of the wind direction (Ekman spiral). However, since the plumes a very short and likely meandering, it is very difficult to estimate the wind direction from the true color composite alone in Figure A5a. Nonetheless, I think the wind direction at the cooling tower is more southerly than indicated by the blue arrows. To conclude, the authors need to add more support for their claim that H2O has been detected from the cooling towers.

**Response:** This is a very astute observation and the authors agree that the observed H2O signal could be a combination of both H2O from the cooling towers in addition to the cooling ponds. This distinction was made on page 14, line 35-36 as well as in the abstract.

In summary, I recommend publication of this paper in AMT after adding the points above as well as the following specific comments and technical corrections.

**Response:** We appreciate the reviewer's detailed assessment of this work and constructive comments.

Specific comments

• Adding a map showing all measurements sites to the paper would make it easier to locate the sites.

**Response:** We agree with the reviewer and have added a new figure showing the locations of the sites (see page 4, Figure 2).

• The authors should change their units from ppm m to a more common unit such as dry air column averaged mole fractions (XCO2 and XCH4).

**Response:** We determined that it would be best to use a colorscale showing units in both ppm-m and the gas state vector at the last iteration of the retrieval (gas scaling factor). This ensures consistency with a number of previous studies (Thorpe et al., 2014, 2016; Thompson et al., 2015, 2016; Frankenberg et al., 2016).

Technical corrections

• Page 6, Section 5.1, 3rd paragraph: Add full stop after "east edge of the AVIRISNG scene"

**Response:**  This typo was corrected (see page 11, line 15).

• Page 7, line 39: Frankenberg et al. (2016) -> (Frankenberg et al. 2016)

**Response:** We could not find this at the stated location (page 7, line 39), but we believe the reviewer was referring to page 12, line 29-30, where the correction was made two times.

[revised manuscript text omitted]

The covariance $\hat{S}$ was calculated to estimate expected IMAP-DOAS retrieval errors as follows:

$$\hat{S} = \left(\mathbf{K}^T \mathbf{S}_\varepsilon^{-1} \mathbf{K} + \mathbf{S}_a^{-1}\right)^{-1} \tag{4}$$

where the diagonal of $\hat{S}$ corresponds to the covariance at each atmospheric layer associated with the gases used for each fitting window. $\mathbf{S}_\varepsilon$, the error covariance matrix, is a diagonal matrix representing expected errors for the retrieval algorithm. For each gas retrieval, the square root of the corresponding diagonal entry of $\hat{S}$ is multiplied by the VMR in the lowest layer of the atmospheric model for each retrieved gas ($CH_4$: 1.86 ppm, $CO_2$: 399 ppm, $H_2O$: 7,745 ppm). Using scene parameters for a 1 km flight altitude above sea level with 25.6° solar zenith and variable signal to noise ratio, this corresponds to an error of

between 0.14 and 0.55 ppm $CH_4$ beneath the aircraft. For $CO_2$, the error ranges between 6.6 to 26.4 ppm and for $H_2O$ between 9.4 to 37.5 ppm.

**5   Results**

**5.1   $CH_4$ emissions from natural gas sector**

AVIRIS-NG identified over 250 $CH_4$ plumes during the Four Corners flight campaign (Frankenberg et al., 2016) using a linearized matched filter (Thompson et al., 2015b). The linearized matched filter models the background of radiance spectra as a multivariate Gaussian and provides a scalar value that represents the fraction of the gas target signature that perturbs the background. Because the target signature is defined as the change in radiance of the background caused by adding a unit mixing ratio length of $CH_4$, detected quantities are reported in mixing ratio lengths (ppm-m). This method is computationally efficient, accounts for the full covariance of background (atmosphere and surface) and instrument noise using in-scene data, providing high sensitivity to local enhancements.

The current speed of the IMAP-DOAS retrieval algorithm precludes it from being applied to all 250 examples presented in the previous study (Frankenberg et al., 2016). Instead, 
[revised manuscript text omitted]
, an $H_2O$ plume is also visible (Figure 12d) emanating from a region that contains a number of cooling towers adjacent to two large cooling ponds (Figure 13a). $CH_4$ retrieval results are also shown in Figure 13c indicating $CH_4$ plumes are not visible in the scene and emphasizing the ability of these retrievals to distinguish between $CH_4$ and $H_2O$ despite spectral interference (see Figure 1). Results for dark surfaces like the cooling ponds were removed from Figure 12b by excluding radiances less than 0.10 $\mu$Wcm$^{-2}$ sr$^{-1}$ nm$^{-1}$ for any band of the $CO_2$ fitting window, for radiances less than 0.002 $\mu$Wcm$^{-2}$ sr$^{-1}$ nm$^{-1}$ for any

[Figure]

**Figure 8.** (a) AVIRIS-NG measured and modeled radiance for one image pixel within the $CO_2$ plume for the $CO_2$ retrieval (see Figure 7b). (b) The residual is plotted with $1\,\sigma$ standard deviation boundary calculated from residuals for the entire scene.

band of the $H_2O$ fitting window (Figure 12d), and for radiances less than 0.01 $\mu Wcm^{-2}\,sr^{-1}\,nm^{-1}$ for any band of the $CH_4$ fitting window (Figure 12c).

In Figure 13a, the AVIRIS-NG true color image is shown for the close up indicated by the black box in Figure 12. The flue-gas stacks are visible in the lower left as $CO_2$ sources and cooling towers in the upper right as possible $H_2O$ sources. Ellipses delineate the shapes of plumes visible in the true color images for the flue-gas stacks (red) and cooling towers (blue). The arrows indicate winds to the southeast for the flue-gas stacks (consistent with $CO_2$ plumes in Figure 12b) and to the east for the cooling towers (consistent with $H_2O$ plumes in Figure 12d). In 13b, the higher resolution Google Earth imagery clearly indicates the flue-gas stacks are much taller (182 m) than the cooling tower (TRI, 2016) based on assessment of shadows, which could explain variable wind directions at the flue-gas stacks and in the vicinity of the cooling towers. Given the presence of the cooling ponds immediately adjacent to the cooling towers, it is unclear if the observed $H_2O$ plume shown in Figure 12d is caused solely by the cooling towers or reflects the combined influence of the towers and evaporation from the cooling ponds.

[revised manuscript text omitted]